# Entanglement Rényi negativity of interacting fermions from quantum Monte Carlo simulations

Fo-Hong Wang [1] & Xiao Yan Xu [1,2] ✉

Many-body entanglement unveils additional aspects of quantum matter and offers insights into strongly correlated physics. While ground-state entanglement has received much attention in the past decade, the study of mixed-state quantum entanglement using negativity in interacting fermionic systems remains largely unexplored. We demonstrate that the partially transposed density matrix of interacting fermions, similar to their reduced density matrix, can be expressed as a weighted sum of Gaussian states describing free fermions, enabling the calculation of rank-*n* Rényi negativity within the determinant quantum Monte Carlo framework. We calculate the rank-two Rényi negativity for the half-filled Hubbard model and the spinless *t-V* model. Our calculation reveals that the area law coefficient of the Rényi negativity for the spinless *t-V* model has a logarithmic finite-size scaling at the finite-temperature transition point. Our work contributes to the calculation of entanglement and sets the stage for future studies on quantum entanglement in various fermionic many-body mixed states.

The characterization of emerging quantum many-body phenomena is multifaceted. Traditionally, physicists have relied on local measurements based on linear response to investigate matter. In recent decades, the utilization of quantum entanglement, a fundamental concept in quantum physics and a powerful tool in quantum information, has become pivotal in unveiling the additional aspects of quantum matter, including the identification of exotic phases and quantum criticality[1–3]. A prominent example is the entanglement entropy (EE) used in bipartite ground-state entanglement studies[4], where various corrections to the area law[5] have been employed to classify quantum phases. These include logarithmic corrections in the leading area-law term for 1D critical systems[6] and Fermi surfaces in generic dimensions[7], subleading logarithmic terms for corner contributions to 2D critical systems[8] and Goldstone modes in symmetry-breaking phases[9], and the topological EE for non-local orders[10,11].

However, EE is not a faithful mixed-state entanglement measurement due to its incompetence in distinguishing quantum entanglement from classical correlation. Thus, many entanglement measurements for mixed states have been proposed[12], including the

entanglement negativity[13–16] (referred to as "negativity" henceforth for brevity), which was designed base on positive partial transpose criteria for the separability of density matrices[17,18]. The evaluation of negativity hinges on the partial transpose of the given density matrix and can be carried out straightforwardly through basic matrix manipulations without invoking any optimization. Hence, negativity has been employed to examine entanglement in finite-temperature Gibbs states or tripartite ground states in various systems, spanning from one-dimensional conformal field theory[19–21], bosonic systems[22–25], spin systems[26–30], to topologically ordered phases[31–35].

In the case of fermionic systems, the definition of partial transpose needs to be adjusted to accommodate the anticommuting statistical property. There exist two different proposals for fermionic partial transpose (FPT) and corresponding fermionic negativity, as discussed in refs. 36,37 and refs. 38–40 respectively. Despite being a computable entanglement measurement, fermionic negativity is only analytically tractable in free systems, especially at finite temperatures, and there have been studies based on both the former definition[25,41–43] and the latter definition[38,44,45]. Therefore, it is desirable to design a

[1]Key Laboratory of Artificial Structures and Quantum Control (Ministry of Education), School of Physics and Astronomy & Tsung-Dao Lee Institute, Shanghai Jiao Tong University, Shanghai 200240, China. [2]Hefei National Laboratory, Hefei 230088, China. ✉e-mail: xiaoyanxu@sjtu.edu.cn

quantum Monte Carlo (QMC) algorithm for large-scale simulation of interacting fermionic systems in an unbiased manner, which is the main goal of this work. Throughout this paper, we adopt the definition in refs. 38,39 under which the partial transpose of a Gaussian state remains a Gaussian state. Additionally, instead of utilizing the originally proposed negativity which involves trace norm of partially transposed density matrices (PTDMs)[15], we consider Rényi negativity (RN) which involves moments of PTDMs, as done in several previous studies on other systems[19,20,23,25,28,30].

In fact, our main result is more broadly applicable. We show that generic PTDMs can be written as a weighted sum of Gaussian states, representing free fermions coupled with auxiliary fields, similar to Grover's pioneering work on reduced density matrices for EE[46]. Our finding facilitates the calculation of RN in a tractable manner, thus establishing it as a powerful tool for characterizing entanglement in mixed states of interacting fermions. We demonstrate this relation using determinant quantum Monte Carlo (DQMC) simulations[47–49] on two paradigmatic models in the realm of strongly correlated electrons, namely, the Hubbard model and the spinless $t$-$V$ model. These two models on bipartite lattices at half-filling are sign-problem-free and both ground-state and finite-temperature properties can be feasibly simulated within the DQMC framework. The relation between negativity and finite temperature transition in fermionic systems is unveiled.

## Results

### Partially transposed density matrix integrated to DQMC framework

Various definitions of negativity in the literature share a common and central dependency, namely, the partial transpose of the density matrix. In this work, we adopt the partial time-reversal transformation proposed by Shapourian et al.[38,39] as the FPT.

We begin with the general partitioning of a fermionic lattice model. It is defined using annihilation (creation) operators $c_{j\sigma}^{(\dagger)}$, which satisfy the anticommutation relations $\{c_{j\sigma}, c_{k\sigma'}^\dagger\} = \delta_{jk}\delta_{\sigma\sigma'}$, where $j, k = 1, \ldots, N$ are the labels of the sites and $\sigma, \sigma'$ are the indices for internal degrees of freedom such as spin. In the following discussion, we may use a column vector $\mathbf{c} = (c_{1,\uparrow}, \ldots, c_{N,\uparrow}, c_{1,\downarrow}, \ldots, c_{N,\downarrow})^T$ to compactly encapsulate all the fermionic operators. This lattice system, denoted as $A$, generally exists within a larger space, as illustrated in Fig. 1a. After tracing out the environment $\bar{A}$, system $A$ typically exists in a mixed state $\rho$. For example, if system $A$ is in contact with a much larger thermal bath at temperature $T$, then we obtain a finite-temperature Gibbs state $\rho = e^{-\beta H}/\text{Tr} e^{-\beta H}$ with $\beta = 1/T$ the inverse temperature and $H$ the Hamiltonian of the system $A$. Next, we further divide system $A$ into two parties belonging to two complementary spatial regions respectively, i.e., $A = A_1 \cup A_2$. Then the density matrix acting on Hilbert space $\mathcal{H}_1 \otimes \mathcal{H}_2$ can be expanded as $\rho = \sum_{A_1,A_2,A_1',A_2'} \rho_{A_1,A_2:A_1',A_2'} |A_1\rangle|A_2\rangle\langle A_1'|\langle A_2'|$.

The FPT of density matrix $\rho$ with respect to subsystem $A_2$, denoted as $\rho^{T_2^f}$, exhibits a highly succinct mathematical expression in the Majorana basis[38,39]. Under Majorana basis, an arbitrary density operator can be expressed as a constrained superposition of products of Majorana operators, which are defined as $\gamma_{2j-1,\sigma} = c_{j,\sigma} + c_{j,\sigma}^\dagger$ and $\gamma_{2j,\sigma} = -i(c_{j,\sigma} - c_{j,\sigma}^\dagger)$. It is found that $\rho^{T_2^f}$ can be obtained by applying the following transformation to the Majorana operators associated with subsystem $A_2$:

$$\mathcal{R}_2^f(\gamma_{j,\sigma}) = i\gamma_{j,\sigma}, \quad j \in A_2. \tag{1}$$

Remarkably, under this definition, the partial transpose of a Gaussian state, denoted as $\rho_0 \sim e^{\frac{1}{4}\boldsymbol{\gamma}^T W \boldsymbol{\gamma}}$ with $\boldsymbol{\gamma} = (\gamma_{1,\uparrow}, \ldots, \gamma_{2N,\uparrow}, \gamma_{1,\downarrow}, \ldots, \gamma_{2N,\downarrow})^T$, retains its Gaussian nature. The question then pertains to determining the explicit form of $\rho_0^{T_2^f}$ or $W^{T_2^f}$. To this end, it is important to emphasize that a Gaussian state $\rho_0$ can be alternatively characterized

**Fig. 1 | A schematic illustration of the core concepts of this work. a** Illustration of the general tripartite geometry. First, we trace out the environment $\bar{A}$ to obtain the reduced density matrix $\rho_A$, and then evaluate the entanglement between subsystems by either tracing out or partially transposing $A_2$. **b** The fermionic partial transpose of Gaussian states, $\rho_0[G] \sim e^{\mathbf{c}^\dagger \ln(G^{-1}-I)\mathbf{c}}$, remains Gaussian. These states represent free fermions, including the auxiliary-field-dependent density matrix $\rho_{\mathbf{s}}$ in the DQMC framework. The reduced and partially transposed density matrices thus share a unified expression in terms of the corresponding Green's functions. Specifically, the reduced Green's function is $G_{\mathbf{s},A_1}^{\text{red}} = \langle c_j c_k^\dagger\rangle_{\mathbf{s}}$ for $j, k \in A_1$, while the partially transposed Green's function $G_{\mathbf{s},A_1}^{\text{FPT}} = G_{\mathbf{s}}^{T_2^f}$ is given by Eq. (3).

by the Green's function $\Gamma_{kl} = \langle[\gamma_k, \gamma_l]\rangle/2$, which is averaged with respect to $\rho_0$ itself and also called covariance matrix.

This matrix is connected to the $W$ matrix through the relation $\tanh(-W/2) = \Gamma$, or inversely, $W = \ln\left[(I+\Gamma)^{-1}(I-\Gamma)\right]$[50] (see also the Supplementary Note 3 for a proof). By employing the definition of $\Gamma$ and the partial transpose in the Majorana basis (refer to Eq. (1)), the partial transpose of the covariance matrix can be formulated as

$$\Gamma^{T_2^f} = \begin{pmatrix} \Gamma^{11} & i\Gamma^{12} \\ i\Gamma^{21} & -\Gamma^{22} \end{pmatrix}, \tag{2}$$

where $\Gamma^{bb'}$ ($b, b' = 1, 2$) denotes the block comprising the matrix elements with rows pertaining to subsystem $A_b$ and columns pertaining to subsystem $A_{b'}$. The Gaussian state described by $\Gamma^{T_2^f}$ precisely yields $\rho_0^{T_2^f}$ through the relation $\tanh(-W^{T_2^f}/2) = \Gamma^{T_2^f}$, i.e., $(\rho_0[\Gamma])^{T_2^f} = \rho_0[\Gamma^{T_2^f}]$ with $\rho_0[\Gamma] \sim e^{\boldsymbol{\gamma}^T \ln[(I+\Gamma)^{-1}(I-\Gamma)]\boldsymbol{\gamma}}$. This elegant fact is proved using Wick's theorem for Majorana monomials[36] (see the Supplementary Note 3 for details).

The above discussion in the Majorana basis can be seamlessly transitioned to the complex fermion basis. In complex fermion basis, the Green's function is defined as $G_{jk} = \langle c_j c_k^\dagger\rangle$, where we have abbreviated the spin indices. Its partially transposed form exhibits also a simple structure

$$G^{T_2^f} = \begin{pmatrix} G^{11} & iG^{12} \\ iG^{21} & I - G^{22} \end{pmatrix}, \tag{3}$$

where the superscripts of the blocks $G^{bb'}$ indicate the subsystems, akin to the notation of $\Gamma^{bb'}$ established earlier. Similar to the Majorana basis, the above Green's function delineates another Gaussian state which is exactly the partial transpose of the original Gaussian state, i.e., $(\rho_0[G])^{T_2^f} = \rho_0[G^{T_2^f}]$ with $\rho_0[G] \sim e^{\mathbf{c}^\dagger \ln(G^{-1}-I)\mathbf{c}}$[51].

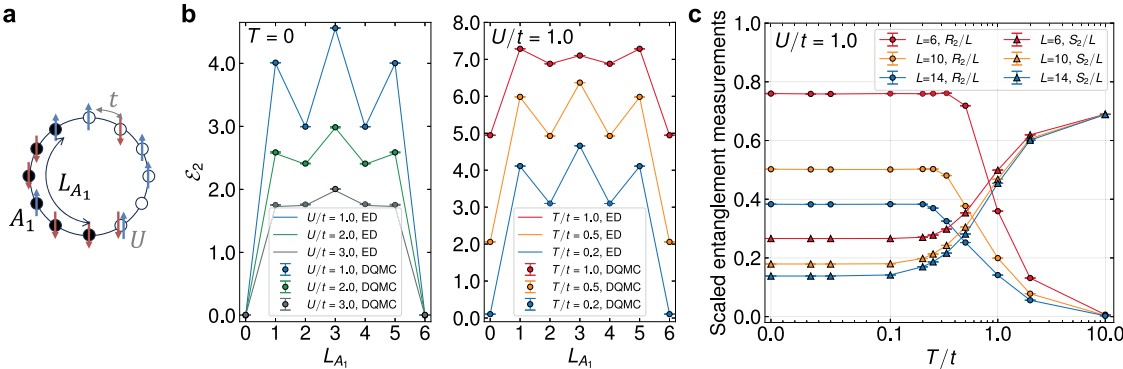

**Fig. 2 | Rank-two RN(R) in the half-filled Hubbard chain. a** A schematic illustration of the half-filled Hubbard chain with periodic boundary conditions and its bipartition geometry. **b** The variation of the rank-two RN $\mathcal{E}_2$ for a six-site Hubbard chain is depicted as a function of the subsystem length $L_{A_1}$. The solid lines represent the ED results, which agree with the DQMC results at both zero temperature (left panel) and finite temperatures (right panel). **c** Quantum-classical crossover. The scaled RNR $R_2/L$ and EE $S_2/L$ of the half-filled Hubbard chain under a half-chain bipartition (i.e., we take $L_{A_1} = L/2$) vary as functions of temperature. As the temperature rises, the scaled EE for different lengths increases and converges, indicating a dominance of volume law at high temperatures. Meanwhile, the RNR begins to vanish once the temperature reaches a critical value associated with the finite-size gap $1/L$[44]. The error bars in (**b**) and (**c**) represent the standard errors from Monte Carlo sampling.

It is now pertinent to redirect our attention towards the partial transpose for interacting fermionic systems, whose density matrices are not Gaussian states. Nonetheless, within the framework of DQMC, the original Hamiltonian $H$ is transformed by replacing interaction terms with fermion bilinears coupled to spacetime-dependent auxiliary fields **s**[47–49]. Specifically, we consider the Gibbs state $\rho = e^{-\beta H}/Z$, where $H$ consists of a free-fermion term $H_0$ and a two-particle interaction term $H_I$. We employ Trotter decomposition to factorize the exponential operator $e^{-\beta H}$ as $(e^{-\Delta_\tau H})^{L_\tau} = \prod_{l=1}^{L_\tau} e^{-\Delta_\tau H_I} e^{-\Delta_\tau H_0} + O(\Delta_\tau^2)$ with $L_\tau = \beta/\Delta_\tau$ being the number of imaginary-time slices. We then apply a Hubbard-Stratonovich (HS) transformation to decouple the interaction terms across different time slices. This procedure yields

$$\rho = \frac{1}{Z}\sum_{\mathbf{s}} \eta[\mathbf{s}] \prod_{l=1}^{L_\tau} \left( e^{\mathbf{c}^\dagger V[\mathbf{s}(l)]\mathbf{c}} e^{-\Delta_\tau H_0} \right) \equiv \sum_{\mathbf{s}} P_\mathbf{s}\rho_\mathbf{s}, \quad (4)$$

where each **s**-configuration is distributed over both the imaginary-time and spatial directions, contributing a scalar factor $\eta[\mathbf{s}]$ and a product of Gaussian operators. Here, both $V[\mathbf{s}]$ and $\eta[\mathbf{s}]$ are derived from $H_I$, and their forms depend on the specific interactions and HS decoupling channels. For detailed expressions related to the two models examined in this study, please refer to the Supplementary Note 2. Since the product of Gaussian states remains a Gaussian state up to a normalization factor[50], the interacting fermionic density matrix $\rho \sim e^{-\beta H}$ can ultimately be written as a weighted sum of Gaussian operators $\rho_\mathbf{s}$, with $P_\mathbf{s}$ denoting the configuration weight[46]. By leveraging the linearity of the partial transpose, we can first individually compute the FPT for each Gaussian state $\rho_\mathbf{s}$. We then sum these results, weighted by their respective probabilities $P_\mathbf{s}$, to obtain the FPT of the entire density matrix $\rho$:

$$\rho^{T_2^f} = \sum_{\mathbf{s}} P_\mathbf{s}\rho_\mathbf{s}^{T_2^f}, \quad (5)$$

where

$$\rho_\mathbf{s}^{T_2^f} = \det\left[G_\mathbf{s}^{T_2^f}\right] \exp\left\{ \mathbf{c}^\dagger \ln\left[ \left(G_\mathbf{s}^{T_2^f}\right)^{-1} - I \right] \mathbf{c} \right\}. \quad (6)$$

The aforementioned equations (5) and (6), along with Eq. (3), are the main results of this work and can be employed to investigate negativity

and negativity spectrum within the conventional DQMC framework, fully analogous to the analysis of EE and entanglement spectrum (see Fig. 1b), respectively.

## Quantum-classical crossover in Hubbard chain

We first consider the half-filled Hubbard chain with periodic boundary conditions, illustrated in Fig. 2a and described by the Hamiltonian

$$H = -t\sum_{\langle ij\rangle\sigma}(c_{i\sigma}^\dagger c_{j\sigma} + \text{H.c.}) + \frac{U}{2}\sum_i (n_i - 1)^2, \quad (7)$$

which is a sign-problem-free model[49]. We will benchmark DQMC results from two perspectives: (i) a numerical comparison with results obtained from exact diagonalization (ED) where we employ the definition of FPT in the Fock space (see Supplementary Eqs. (2) and (3)), and (ii) providing a physical explanation for why the negativity is a more competent mixed-state entanglement measurement compared to EE in the context of a quantum-classical crossover[38,44].

We define the rank-$n$ RN as

$$\mathcal{E}_n = -\frac{1}{n-1}\ln\text{Tr}\left[\left(\rho^{T_2^f}\right)^n\right], \quad (8)$$

where the $n$-th moment of the PTDM, denoted as $\text{Tr}[(\rho^{T_2^f})^n]$, is also referred to as the replica approach of negativity in previous studies[19,20]. The quantity $\mathcal{E}_n$ is formally a direct analog to rank-$n$ Rényi EE $S_n(A_1) = -(\ln\text{Tr}\rho_{A_1}^n)/(n-1)$, where $\rho_{A_1} = \text{Tr}_{A_2}\rho$ represents the reduced density operator obtained after tracing out subsystem $A_2$. Utilizing Eq. (5), we can derive the DQMC expression for measuring, for instance, the rank-two RN

$$\mathcal{E}_2 = -\ln\left\{ \sum_{\mathbf{s}_1\mathbf{s}_2} P_{\mathbf{s}_1} P_{\mathbf{s}_2} \det\left[ G_{\mathbf{s}_1}^{T_2^f} G_{\mathbf{s}_2}^{T_2^f} + \left(I - G_{\mathbf{s}_1}^{T_2^f}\right)\left(I - G_{\mathbf{s}_2}^{T_2^f}\right) \right] \right\}. \quad (9)$$

The distinction between the FPT and the conventional one is presented herein. For bosonic systems, considering that $\text{Tr}[(\rho^{T_2})^2] = \text{Tr}[\rho^2]$, the rank-two RN becomes trivial, thereby rendering the minimal meaningful rank as three[19,20,30]. However, we show in the occupation number representation that all fermionic PTDMs satisfy $\text{Tr}[(\rho^{T_2^f})^2] = \text{Tr}[(\rho\hat{X}_2(\pi))^2]$ with $\hat{X}_2(\theta) = e^{i\theta\sum_{j\in A_2} n_j}$ being the disorder operator (see the Supplementary Note 1 for details). Consequently, $\mathcal{E}_2$ can reveal the

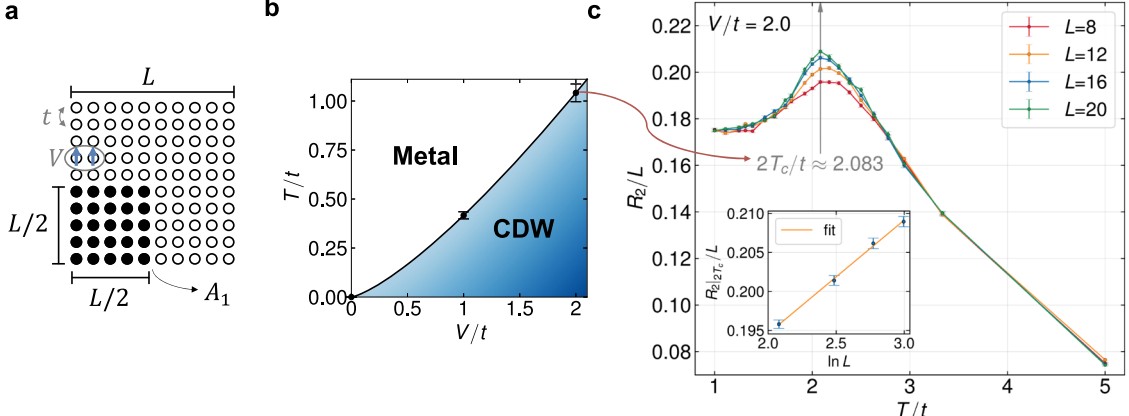

**Fig. 3 | Rank-two RNR in the *t-V* model across a finite-temperature transition point. a** A schematic illustration of the spinless (or spin-polarized) *t-V* model on a square lattice and the chosen one-quarter bipartition geometry. **b** The phase diagram of the *t-V* model, where the phase boundary points at $V = 1$ and $V = 2$ are determined through mixed-state entanglement studies. One can find the data of $V = 1$ case in the Supplementary Fig. 2. The error bars for these two points are estimated based on the difference between the neighboring temperature points that were calculated. **c** The finite-temperature transition in the spinless *t-V* model is detected by the area-law coefficient of the RNR as a function of temperature. A gray vertical arrow indicates the position of the shared peak, with half of its magnitude aligning with the transition point determined in previous studies[53,55]. The inset shows the linear scaling of the area-law coefficient at the critical point with $\ln L$. The error bars in (**c**) represent the standard errors from Monte Carlo sampling.

entanglement information of the system. As shown in Fig. 2b, the results calculated by DQMC and ED show strong agreement in both the zero-temperature and the finite-temperature regimes. In the former regime, the pattern of the rank-two RN exhibits analogous variations to those of the rank-two Rényi EE[46,52], in response to alterations in the length of subsystem $A_1$, denoted as $L_{A_1}$. However, at finite temperatures, the negativity maintains a symmetric pattern, which is different from the behavior of EE[52,53]. As the temperature rises, the magnitude of the negativity increases, resulting in an overall non-zero shift corresponding to a non-zero thermodynamic entropy of $-\ln(\mathrm{Tr}\rho^2)$.

Based on the above observation at finite temperatures, we also examine the ratio between $\mathrm{Tr}[(\rho^{T_2^f})^n]$ and $\mathrm{Tr}[\rho^n]$ dubbed the Rényi negativity ratio (RNR)[23,28,30]

$$R_n = -\frac{1}{n-1}\ln\left\{\frac{\mathrm{Tr}\left[\left(\rho^{T_2^f}\right)^n\right]}{\mathrm{Tr}[\rho^n]}\right\} = \mathcal{E}_n - S_n^{\mathrm{th}}, \tag{10}$$

where $S_n^{\mathrm{th}} = -(\ln\mathrm{Tr}\rho^n)/(n-1)$ denotes the thermodynamic Rényi entropy, which equals $\mathcal{E}_n$ for either $A_1 = A$ or $A_2 = A$. A faithful description of mixed-state entanglement necessitates the exclusion of the thermodynamic Rényi entropy $S_n^{\mathrm{th}}$. In Fig. 2c, we display the variations of the RNR and EE with temperature for three distinct lengths, namely $L = 6, 10, 14$. Here, the subsystem $A_1$ is chosen to be half of the chain, yielding an equal bipartition. As the temperature rises, the EE increases while the RNR asymptotically diminishes to zero for all lengths. This serves as a compelling physical demonstration of the quantity $R_n$. In a generic mixed state, both quantum and classical correlations are present, and effective measurement of mixed-state entanglement should exclusively isolate the quantum correlations[15]. In the specific context of finite-temperature Gibbs states, the classical correlation is simply the thermal fluctuations delineated by the thermodynamic entropy $S_n^{\mathrm{th}}$. Furthermore, at sufficiently low temperatures, the RNR remains constant and establishes a plateau, the length of which is associated with the finite-size gap $1/L$[44]. As depicted in Fig. 2c, it is evident that with an increase in chain length, the plateau becomes narrower. In summary, the monotonic decay of the RNR with rising temperature signifies a crossover from a quantum entangled state to a classical mixed state.

## Finite temperature transition in *t-V* model

To demonstrate the efficacy of the RNR in detecting finite-temperature phase transition, we further consider the half-filled spinless *t-V* model on a square lattice with periodic boundary conditions[53–55],

$$H = -t\sum_{\langle i,j\rangle}(c_i^\dagger c_j + c_j^\dagger c_i) + V\sum_{\langle i,j\rangle}\left(n_i - \frac{1}{2}\right)\left(n_j - \frac{1}{2}\right), \tag{11}$$

where both the hopping and the interaction involve only nearest neighbors (see Fig. 3a for illustration). In the presence of a finite coupling parameter $V$, this model exhibits a charge density wave (CDW) ground state and undergoes a phase transition from the CDW phase to a metallic phase at finite temperature (see Fig. 3b), with critical behavior falling within the 2D Ising universality class[55,56]. In the following, we focus on a specific coupling strength, $V/t = 2$, where the critical temperature was estimated to be $T_c/t \approx 1.0$[55].

This model is also a sign-problem-free model[57–60]. However, for models with larger dimensions or stronger interaction strengths, the direct sampling of RN using Eq. (9) becomes inaccurate, as a result of the occurrence of spikes[61] or the non-Gaussian distribution of Grover determinants $\det g_x = \det[G_{s_2}^{T_2^f} G_{s_2}^{T_2^f} + (I - G_{s_1}^{T_2^f})(I - G_{s_2}^{T_2^f})]$[62]. We implement an incremental algorithm for the RN, analogous to the controllable incremental algorithm for EE[62–64], the spirit of which is to measure $(\det g_x)^{1/N_{\mathrm{inc}}}$ instead of $(\det g_x)$ to circumvent the sampling of an exponentially small quantity with exponentially large variance (see "Methods"). It is important to note that there is a sign ambiguity in the $N_{\mathrm{inc}}$-th root. In the Supplementary Note 5, we prove that the Grover determinant $\det g_x$ is always real and non-negative for two classes of sign-free models, represented by the Hubbard model and the spinless *t-V* model, respectively.

As illustrated in Fig. 3a, we designate the lower left corner with dimensions $(L/2) \times (L/2)$ as subsystem $A_1$, resulting in an area-law coefficient of the RNR of $R_2/L$. The main plot of Fig. 3c presents $R_2/L$ as a function of temperature for various system sizes, demonstrating a notably distinct finite-size characteristic compared to the intersection of mutual information[53,65]. Remarkably, unlike the Hubbard model in Fig. 2c or the previous study on the 2 + 1D transverse field Ising model[30], the RNR does not exhibit a monotonic decrease with rising temperature. Instead, for varying lattice sizes, a shared local maximum

appears at approximately twice the transition temperature, $2T_c/t \approx 2.1$. The inclusion of the prefactor 2 aligns with the rank of the RNR under consideration, consistent with earlier discussion on the critical behavior within replica approach[30,33,35]. In general, the singularity of $R_n$ is anticipated to occur at $T = nT_c$ with $T_c$ being the physical transition temperature. This expectation arises because $R_n$, as indicated by the denominator in its definition in Eq. (10), i.e., $\text{Tr}[e^{-n\beta H}]$, effectively corresponds to a Gibbs state with an effective inverse temperature of $n\beta$. Therefore, we demonstrate that it is possible to quantitatively extract the finite-temperature transition points and the phase diagram of the $t$-$V$ model (see Fig. 3B) from mixed-state entanglement studies. Finally, we briefly highlight the finite-size scaling of the rank-two RNR. As shown in the inset of Fig. 3c, the peaks of $R_2/L$ exhibit a logarithmic divergence with system size $L$, while the area law is well-preserved in regions far from the critical point. This beyond-area-law scaling around the finite temperature critical point is also observed for other values of $V$ (namely, $V = 0$ and $V = 1$), a different bipartite geometry, and a different lattice. Refer to the Supplementary Fig. 2 for additional complementary plots.

## Discussion

We showed that the PTDM for interacting fermions, akin to the reduced density matrix, can be expanded as a weighted sum of Gaussian states representing free fermions, thereby paving the way for the study of mixed-state entanglement in strongly correlated fermionic systems. This main result was employed to implement an algorithm to compute the rank-$n$ RN for interacting fermionic systems within the DQMC framework. We studied the rank-two RN for the half-filled Hubbard chain and the spinless $t$-$V$ model on a square lattice. Remarkably, we found that the area law coefficient of the RNR exhibits a logarithmic singular peak at about twice the finite-temperature transition point for all lattice sizes under consideration.

We now discuss the possible physical interpretations of the logarithmic divergence of the rank-two RNR at $2T_c$. Based on symmetry considerations, it was argued that the entanglement negativity inherits the singularity of the specific heat at a finite temperature transition[30,35], and for the 2D Ising transition, the specific heat has a logarithmic divergence in lattice linear size. However, in our fermionic scenario, the quantity showing this divergence is $R_2/L$ rather than its temperature derivative. Thus it can not be directly connected to specific heat and the underlying cause of the logarithmic divergence of $R_2/L$ at $2T_c$ remains an open issue. We note that refs. 30,35 concerning bosonic models used the conventional partial transpose, which may partly account for the inconsistency.

There are several potential future research directions to consider. The first direction is to investigate the finite-temperature entanglement of various interacting fermionic models, especially those with transition points that belong to different universality classes, such as the 3D Hubbard model which owns a transition belonging to O(3) universality class[66]. Further, exploring the entanglement in other types of mixed states, such as tripartite ground states of topological[39] and gapless systems[25], and measurement-induced mixed states[67] presents an intriguing avenue for further research. Next, exploring the finite-size scaling laws of negativity in interacting fermionic systems could also be intriguing. In particular, our finding of the $L \ln L$ scaling of the RNR at the critical point in the spinless $t$-$V$ model may indicate long-range entanglement contribution[30,68], which warrants further investigation. Moreover, our results are applicable to the continuous-time QMC method, offering an opportunity to study the mixed-state entanglement of realistic correlated materials through combining with dynamical mean-field theory[69–73]. In the hybridization expansion algorithm, the bath can be firstly traced out[46,51], allowing the impurity's reduced density matrix to be derived from the Green's functions and density correlation functions[68]. Additionally, the rank-two RN can be

computed within the interaction expansion algorithm via the identity $\text{Tr}[(\rho^{T_2^f})^2] = \text{Tr}[(\rho X_2(\pi))^2]$. Here, the disorder operator $X_2(\tau) \propto \prod_{j \in A_2, \sigma}(n_{j\sigma}(\tau) - \frac{1}{2})$ in the interaction picture introduces additional interaction vertices exclusively in the $A_2$ subregion. Finally, another proposal for realistic materials is to integrate the RN into the constrained-path auxiliary-field QMC method[74–77] which controls the sign problem.

## Methods

### Determinantal quantum Monte Carlo

We used determinantal qauntum Monte Carlo (QMC) to simulate the two interacting fermionic models, both in zero-temperature regime (i.e., projective QMC) and in finite-temperature regime. Interesting readers may refer to the supplementary materials for all the details including basic formalism and the Hubbard-Stratonovich transformations used in this work. The projective DQMC calculations performed in Fig. 2 used a projection length $\Theta/t = 20$, which is long enough to project the trial state to the ground state and ensures desired convergence. We chose the time slice step $\Delta_\tau$ to be between 0.02 and 0.05, depending on the size of $\Theta$ or $\beta$, and the results do not change if we choose a smaller $\Delta_\tau$. The results shown in Fig. 2 of the main text were accelerated by employing the delay update algorithm[78].

### Incremental algorithm for Rényi negativity

The results presented in Fig. 3 were calculated using the incremental algorithm, which was proposed and implemented for Rényi entanglement entropy[62]. We have developed an analogous version for Rényi negativity. Specifically, we measure the exponentially small observable $e^{-(n-1)\mathcal{E}_n}$ by separately calculating its $N_{\text{inc}}$ factors, each of which is of order $O(10^{-1})$,

$$e^{-(n-1)\mathcal{E}_n} = \frac{\sum_{\mathbf{s}} w_{\mathbf{s}_1} \cdots w_{\mathbf{s}_n} \det g_x}{\sum_{\mathbf{s}} w_{\mathbf{s}_1} \cdots w_{\mathbf{s}_n}} = \frac{Z_{N_{\text{inc}}}}{Z_{N_{\text{inc}}-1}} \cdots \frac{Z_{k+1}}{Z_k} \cdots \frac{Z_1}{Z_0}, \quad (12)$$

where $w_{\mathbf{s}}$ is the regular DQMC weight for a specific auxiliary field configuration $\mathbf{s}$. Here, we define intermediate partition functions as $Z_k = \sum_{\mathbf{s}_1 \cdots \mathbf{s}_n} w_{\mathbf{s}_1} \cdots w_{\mathbf{s}_n} (\det g_x)^{\frac{k}{N_{\text{inc}}}}$. Each ratio can be interpreted as the average value of the $N_{\text{inc}}$-th root of the Grover determinant, $(\det g_x)^{1/N_{\text{inc}}}$, over a replicated system with weight $W_{\mathbf{s}_1 \cdots \mathbf{s}_n} = w_{\mathbf{s}_1} \cdots w_{\mathbf{s}_n} (\det g_x)^{\frac{k}{N_{\text{inc}}}}$

$$\frac{Z_{k+1}}{Z_k} = \frac{\sum_{\mathbf{s}_1 \cdots \mathbf{s}_n} w_{\mathbf{s}_1} \cdots w_{\mathbf{s}_n} (\det g_x)^{\frac{k+1}{N_{\text{inc}}}}}{\sum_{\mathbf{s}_1 \cdots \mathbf{s}_n} w_{\mathbf{s}_1} \cdots w_{\mathbf{s}_n} (\det g_x)^{\frac{k}{N_{\text{inc}}}}} = \frac{\sum_{\mathbf{s}_1 \cdots \mathbf{s}_n} W_{\mathbf{s}_1 \cdots \mathbf{s}_n} (\det g_x)^{\frac{1}{N_{\text{inc}}}}}{\sum_{\mathbf{s}_1 \cdots \mathbf{s}_n} W_{\mathbf{s}_1 \cdots \mathbf{s}_n}}. \quad (13)$$

In our calculations, the number of intermediate processes is $N_{\text{inc}} = 64$, which is sufficiently large to achieve desirable statistical accuracy for the lattice sizes considered.

## Data availability

The data that support the findings of this study are provided at https://scidata.sjtu.edu.cn/records/2xm0w-eng25.

## Code availability

The code used in this work is available from the corresponding author upon request.

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

## Acknowledgements

The authors thank Tarun Grover for helpful discussions and comments on the draft. This work was supported by the National Key R&D Program of China (Grant No. 2022YFA1402702, No. 2021YFA1401400), the National Natural Science Foundation of China (Grants No. 12447103, No. 12274289), the Innovation Program for Quantum Science and Technology (under Grant No. 2021ZD0301902), Yangyang Development Fund, and startup funds from SJTU. The computations in this paper were run on the Siyuan-1 and π 2.0 clusters supported by the Center for High Performance Computing at Shanghai Jiao Tong University.

## Author contributions

F.-H.W. and X.Y.X. performed research, analyzed data, and wrote the paper.

## Competing interests

The authors declare no competing interests.
