## [Transparent Peer Review file · Nature Communications]

Entanglement Rényi Negativity of Interacting Fermions from Quantum Monte Carlo Simulations

Corresponding Author: Professor Xiao Yan Xu

Version 0:

Reviewer comments:

Reviewer #1

(Remarks to the Author)

Key results: This paper considers an approach to estimate the quantum entanglement in many-body fermionic systems. Instead of using the entanglement entropy as a measure of entanglement the authors propose to use Rényi negativity within a finite-temperature quantum Monte-Carlo simulation.

Validity: The results seem plausible. These are derived within the main-stream framework of the Hubbard model and the spinless t-V model. I do not find however the author's explanation for the different behavior of the EE and RNR entropies with the temperature to be rather convincing.

Originality and significance: The results presented are a combination of well-known approaches used in condensed-matter physics. I do not see an interdisciplinary impact of the work outside that field.

Data & methodology: I find the presentation to be rather schematic, with an emphasis on the sequence of mathematical considerations which led the authors to building their models. The physical interpretation of the results presented in figures 2 and 3 lacks clarity that would make it intriguing for broader audiences. I believe in any case these results would need further confirmation from others.

Since the authors employ statistical methods (Monte-Carlo) it would be beneficial for the readers to see a general estimation of the errors due to the method (an idea for that is presented in the inset of fig.3c).

Clarity and context: The abstract, introduction and conclusions are readable for experts in the field of theoretical condensed-matter physics.

I believe the paper contains sufficiently new material, but it seems more appropriate for publication in a specialized journal such as e.g. Phys. Rev. B.

Reviewer #2

(Remarks to the Author)

The paper entitled "Entanglement Rényi Negativity of Interacting Fermions from Quantum Monte Carlo Simulations" aims at quantifying quantum correlations in systems of interacting fermions in mixed states. For example, finite-temperature Gibbs states resulting from a coupling to a thermal bath. The authors focus on the Rényi Negativity (RN), which is defined analogously to the Rényi entanglement entropy but with the fermionic partial transpose replacing the partial trace. Their main result is a demonstration that, within the Determinant Quantum Monte Carlo framework (using Trotter decomposition and Hubbard-Stratonovich decoupling), the partially transposed density matrix can be expressed as a weighted sum of Gaussian operators. Authors apply this approach to two systems: the half-filled Hubbard chain and the half-filled spinless t-V model on a square lattice. This enables them to show that, in the latter model, the RN ratio exhibits a logarithmic correction to the area law, suggesting a transition. However, this correction appears at approximately twice the transition temperature.

The paper is interesting and well-written. I have only a few reservations:

1. Authors consider Eqs. (5) and (6) to be the main results of their work. However, the entire derivation is relegated to the Supplemental Material. I believe the manuscript would benefit from including at least a short summary in the main text, explaining how these key results are established. This would also improve the clarity of the main text, as some quantities, like $K_I[s]$ in the partition function, are not introduced in the main text and appear unclear without referring to the Supplemental Material.
2. When pointing out the fact that the logarithmic correction to the area law appears at twice the transition temperature, authors write "The inclusion of the prefactor 2 aligns with the rank of the RNR under consideration, consistent with earlier discussion on the critical behaviour within replica approach." and refer to the literature. Since this result is even highlighted in the abstract, it would strengthen the manuscript to include its explanation, rather than providing a brief statement and a reference to the literature.
3. In the main text, some quantities are either never introduced or are introduced too late. For example, the column vector "gamma" is never introduced, while the column vector "c" is introduced near the partition function, despite its earlier appearance in the density matrix of a Gaussian state. Additionally, there are some errors in the Supplemental Material. For example, H_U and not H_I is introduced in Eq. (S7), indices are mixed in Eq. (S9), et cetera.
4. Authors may consider adding a short description of the Determinant Quantum Monte Carlo framework to the Supplemental Material. While the main expressions are provided in the main text, short explanation of the framework itself would be beneficial.

After these minor corrections, I recommend the manuscript for publication.

Version 1:

Reviewer comments:

Reviewer #1

(Remarks to the Author)

It is my belief that the papers published in Nature Communications are intended to represent important advances in specific fields together with significance for physics. Therefore, in my first review I emphasized this viewpoint rather than going through the formulae in the manuscript. Also, I don't think it is a good idea to question the reviewer's competence as "lack of logical sequence in his/her comments" or a "lack of familiarity with the realm..." as the reviewer doesn't need to be exactly an expert in "quantum many-body computations" to be able to judge whether a paper represents an important advance or not.

Here I'll leave aside the statement in author's response that the entanglement entropy cannot characterize entanglement of mixed state, which is simply wrong. For example, a simple molecule is in a pure state at zero temperature but a single atom within it is in a mixed state and its entanglement can be quantified by entanglement entropy. I'll not also comment on the author's statement that their methods are applicable to "realistic many-body systems" simply because the validity of Hubbard model they employ is by itself a rather crude approximation to the realistic systems as shown previously. Building a new methodology based on an oversimplified model which neglects the detailed electron-electron interactions including the long-range interactions, etc, cannot in my view be considered an important advance. Clearly, benchmarking against exact diagonalization does not help much in this sense.

I think that beyond the technicalities the authors really overestimate the significance of their work saying that it "holds interdisciplinary significance, as it bridges the gap between quantum information theory and condensed matter physics". I don't think so, for I did not see any "new insights into entanglement in strongly correlated systems" or "new avenues" throughout the paper. What I saw are just numerical advances in treating fermionic systems under finite temperature conditions but with no new consequences for physics, no new conclusions, etc. In both the Abstract and the Conclusions, the authors are just saying what they have done. I hope they will finally understand what my point is here... A clear presentation of the limitations of the model, along with clear claims of novelty with respect to new physics, is still missing. Having said all that, I still believe that simply presenting an advance in numerical techniques would not be sufficient to warrant publication in Nature Communications.

Reviewer #2

(Remarks to the Author)

I am satisfied with the modifications to the manuscript made by the authors. Moreover, I stand by my previous assessment that the manuscript is interesting and well-written and, as such, it can be published in Nature Communications. I have only one final suggestion.

I propose that, instead of writing "Nonetheless, within the framework of DQMC, after Trotter decomposition and Hubbard-Stratonovich (HS) decoupling (see Methods and the supplementary materials), the original two-particle interaction terms in the Hamiltonian H are replaced by fermion bilinears coupled with spacetime dependent auxiliary fields s .", the authors could

provide an example. This example could be drawn from the beginning of Supplemental Material S2A, starting with "At a finite temperature T , and assuming that the system of interest is in thermodynamic equilibrium (...)" and ending with "Here, $s(l)$ includes all the auxiliary fields at time slice l and N_c represents the number of coupling terms, which varies depending on the specific interactions and decoupled channels.". Including this example would help visualize the appearance of fermion bilinears while simultaneously establishing a connection with the remainder of the paragraph "Consequently, since the product of Gaussian states remains a Gaussian state (...)."

Response to Nature Communications Submission

Paper Title: Entanglement Rényi negativity of interacting fermions from quantum Monte Carlo simulations

Authors: Fo-Hong Wang and Xiao Yan Xu

Summary of changes made

To sum up, we have made the following changes:

1. We have improved the clarity of the texts around Eqs. (4) and (5) in the main text, as well as the related Section S3 in the supplementary materials, based on the comments from Reviewer 2.
2. As suggested by Reviewer 2, we have added an explanation for why the singular point of the rank- n Rényi negativity ratio occurs at temperature $T = nT_c$ in the section titled “Finite temperature transition in t - V model”.
3. As suggested by Reviewer 2, we have added a brief introduction to the workflows of both finite-temperature and projective determinant quantum Monte Carlo.
4. In the supplementary materials, we have corrected some mistakes that were either pointed out by Reviewer 2 or found by ourselves after careful checking.
5. We have improved the legends of Fig. 2C.

The following is a detailed response to the recommendations and criticisms from the reviewers. The reviewer comments are laid out in *italicized font* and our response is given in normal font.

Reply to Reviewer #1

First of all, we would like to highlight several issues with the report provided by Reviewer #1:

1. The reviewer's comments lack a logical sequence and do not provide a clear rationale for recommending a different journal.
2. Most of the comments are overly general and unclear, making it difficult to discern the specific concerns or intentions behind them.
3. Some criticisms directly reflect a lack of familiarity with the realm of quantum many-body computation and a misunderstanding of our work.

We are committed to improving our manuscript and welcome any specific suggestions for enhancing its clarity and quality. However, the issues listed above make it challenging for us to fully address the concerns raised and improve our manuscript accordingly. In the following, we will respond to the reviewer's comments, point out the unclear or incorrect aspects, and provide our perspective on the evaluations raised.

Reviewer #1: *Key results: This paper considers an approach to estimate the quantum entanglement in many-body fermionic systems. Instead of using the entanglement entropy as a measure of entanglement the authors propose to use Rényi negativity within a finite-temperature quantum Monte-Carlo simulation.*

Our reply: We appreciate the reviewer's interest in our work. While the summary captures one aspect of our contribution, we would like to respectfully clarify and expand upon several key points: (1) Our proposed method is more comprehensive than indicated, as it characterizes entanglement in many-body mixed states across both finite and zero temperatures. This is explicitly demonstrated in Figs. 2b and 2c, which present data points at both finite and zero temperatures. (2) The significance of our approach extends beyond just providing an alternative to entanglement entropy. The Rényi negativity method we developed offers distinct advantages for studying mixed-state entanglement in fermionic systems, while entanglement entropy cannot characterize entanglement of mixed-state. (3) Our quantum Monte Carlo implementation represents a practical computational framework that makes these measurements feasible in realistic many-body systems.

This broader scope of our work is fundamental to its contribution to the field of quantum many-body physics.

Reviewer #1: *Validity: The results seem plausible. These are derived within the main-stream framework of the Hubbard model and the spinless t - V model.*

Our reply: We appreciate the reviewer’s assessment of our results’ plausibility. However, we would respectfully provide further clarification on several points: (1) The comment about “results seem plausible” is quite general. It would be helpful if the reviewer could specify which particular results they are referring to, as our paper presents several key results. (2) Regarding the “main-stream framework” characterization, we find this description potentially ambiguous. While the Hubbard model and spinless t - V model are indeed fundamental models in condensed matter physics, our work’s novelty lies in the development of new methodological approaches to study mixed state entanglement in these systems. We would appreciate if the reviewer could elaborate on what aspects of our framework they consider “main-stream” to help us better address any potential concerns or misunderstandings.

Reviewer #1: *I do not find however the author’s explanation for the different behavior of the EE and RNR entropies with the temperature to be rather convincing.*

Our reply: We appreciate the reviewer’s feedback. We would like to address this comment from two perspectives. (1) The different behaviors of entanglement entropy (EE) and Rényi negativity ratio (RNR) are empirically established through rigorous numerical calculations, benchmarked against exact diagonalization. These distinct behaviors are therefore observed phenomena rather than theoretical interpretations. (2) In Fig. 2c, we provide a detailed analysis of these different behaviors: At high temperatures, the density matrix of the Gibbs state approaches an identity matrix, leading to volume-law scaling of EE (linear increase with system size). For identical model parameters and bipartition geometry, the RNR exhibits contrasting behavior, vanishing at high temperatures. This RNR behavior is consistent with previous findings in free fermionic systems (Ref. [44]). These distinct characteristics suggest that the RNR serves as a more effective probe of mixed-state entanglement compared to EE. Given that our explanation builds upon both numerical evidence and established theoretical frameworks, we would appreciate if the reviewer could specify which aspects of our explanation they find unconvincing. This would help us address any potential concerns more effectively and improve the clarity of our manuscript.

We note a technical inaccuracy in the reviewer’s comment: the term “entropies” following “EE and RNR” is incorrect, as “Entanglement entropy (EE)” already includes the term “entropy”, and the Rényi negativity ratio (RNR) is not classified as an entropy.

Reviewer #1: *Originality and significance: The results presented are a combination of well-known approaches used in condensed-matter physics. I do not see an interdisciplinary impact of the work outside that field.*

Our reply: We thank the reviewer for providing the comments regarding the originality and significance of our work. However, we respectfully disagree with the reviewer’s assessment and would like to offer the following points.

(1) The statement that our results “are a combination of well-known approaches used in condensed-matter physics” is quite vague and does not accurately reflect the novelty of our work. We would appreciate it if the reviewer could specify which “well-known approaches” they are referring to. Our main contribution is the introduction of a new methodological approach to studying mixed-state entanglement in interacting fermionic systems, applicable to a variety of quantum Monte Carlo methods. We have developed a specific computational framework using the determinant quantum Monte Carlo (DQMC) method. While DQMC is indeed a widely used, numerically exact algorithm for studying interacting fermionic systems, our work is the first to integrate it with the RNR to study mixed-state entanglement, significantly extending the applicability of DQMC.

(2) Regarding the interdisciplinary impact of our work, we would like to first emphasize that condensed matter physics is a vast and diverse field encompassing a wide range of topics. Our study examined strongly correlated fermionic systems through the lens of entanglement, a fundamental concept in quantum information theory. Negativity is one of the few computable measures of entanglement for mixed states and is widely utilized in quantum information. Therefore, our work holds interdisciplinary significance, as it bridges the gap between quantum information theory and condensed matter physics.

Reviewer #1: *Data & methodology: I find the presentation to be rather schematic, with an emphasis on the sequence of mathematical considerations which led the authors to building their models.*

Our reply: We appreciate the reviewer’s evaluation. However, the comment that “our presentation is rather schematic” is quite general, making it difficult for us to understand which specific aspects of the presentation may be causing confusion. Additionally, the phrase “the sequence of mathematical considerations” is somewhat ambiguous, and it’s unclear why the reviewer believes it led us to “build the models”.

Our choice of models is primarily based on their sign-problem-free nature, which allows us to perform large-scale quantum Monte Carlo simulations. Furthermore, the Hubbard model and the spinless t - V model are paradigmatic in the study of strongly correlated fermionic systems and have garnered significant interest within the condensed matter community.

Reviewer #1: *The physical interpretation of the results presented in figures 2 and 3 lacks clarity that would make it intriguing for broader audiences. I believe in any case these results would need further confirmation from others.*

Our reply: We appreciate the reviewer’s feedback, but we find it challenging to understand your concerns regarding our physical interpretation of the results. Our paper includes detailed discussions on the physical interpretation of the numerical results, specifically addressing the quantum-classical crossover in Fig. 2 and the

finite-temperature transition in Fig. 3. We would appreciate more specific feedback from the reviewer concerning the clarity of these interpretations. Additionally, the comment that our “results would need further confirmation from others” is quite general and does not provide evidence that our results are incorrect or unreliable.

Our newly proposed algorithm for computing the RNR has been benchmarked against exact diagonalization results for small system sizes (see Fig. 2b), which follows the standard procedure in numerical studies. Furthermore, the large-scale results align with previous studies in certain aspects. Regarding Fig. 2, please refer to our earlier response to the reviewer’s comment (the third point). For Fig. 3, the singular behavior of the RNR at twice the transition temperature is characteristic of the replica trick. This behavior was also compared with another measurement, namely the mutual information, as well as with the RNR of a bosonic model that also exhibits a finite-temperature transition characterized by the same universality.

Reviewer #1: *Since the authors employ statistical methods (Monte-Carlo) it would be beneficial for the readers to see a general estimation of the errors due to the method (an idea for that is presented in the inset of fig.3c).*

Our reply: Not only are the data points in the inset of Fig. 3c accompanied by error bars, but every data point in all figures obtained from quantum Monte Carlo simulations is also presented with error bars.

Reviewer #1: *Clarity and context: The abstract, introduction and conclusions are readable for experts in the field of theoretical condensed-matter physics.*

Our reply: We appreciate the reviewer’s evaluation of the clarity and context provided in our abstract, introduction, and conclusions. If there are any sections outside these areas that may be unclear or require further elaboration—had such specific points been raised—we would be more than willing to refine and enhance them.

Reviewer #1: *I believe the paper contains sufficiently new material, but it seems more appropriate for publication in a specialized journal such as e.g. Phys. Rev. B.*

Our reply: We appreciate the reviewer’s assessment that our work contains sufficiently new material. However, we do not understand and respectfully disagree with the recommendation to submit to another journal. The suggestion lacks specific reasoning, and the unstructured and overly ambiguous nature of the report makes it challenging to interpret. Our work provides an new avenue for numerically studying mixed-state entanglement in various interacting fermionic systems, which applies to a broad range of quantum Monte Carlo methods. We are the first to conduct large-scale quantum Monte Carlo simulations of the Rényi negativity in two paradigmatic models, offering new insights into entanglement in strongly correlated systems. We

believe that our research is particularly well-suited for the broad and interdisciplinary audience of Nature Communications.

Reply to Reviewer #2

Reviewer #2: *The paper entitled “Entanglement Rényi Negativity of Interacting Fermions from Quantum Monte Carlo Simulations” aims at quantifying quantum correlations in systems of interacting fermions in mixed states. For example, finite-temperature Gibbs states resulting from a coupling to a thermal bath. The authors focus on the Rényi Negativity (RN), which is defined analogously to the Rényi entanglement entropy but with the fermionic partial transpose replacing the partial trace. Their main result is a demonstration that, within the Determinant Quantum Monte Carlo framework (using Trotter decomposition and Hubbard-Stratonovich decoupling), the partially transposed density matrix can be expressed as a weighted sum of Gaussian operators. Authors apply this approach to two systems: the half-filled Hubbard chain and the half-filled spinless t - V model on a square lattice. This enables them to show that, in the latter model, the RN ratio exhibits a logarithmic correction to the area law, suggesting a transition. However, this correction appears at approximately twice the transition temperature.*

The paper is interesting and well-written. I have only a few reservations:

Our reply: We sincerely thank the reviewer for the careful reading and positive assessment of our work. In the following, we address the specific points raised by the reviewer.

Reviewer #2: *1. Authors consider Eqs. (5) and (6) to be the main results of their work. However, the entire derivation is relegated to the Supplemental Material. I believe the manuscript would benefit from including at least a short summary in the main text, explaining how these key results are established. This would also improve the clarity of the main text, as some quantities, like $K_l[s]$ in the partition function, are not introduced in the main text and appear unclear without referring to the Supplemental Material.*

Our reply and changes: We thank the reviewer for pointing out this problem. The main results of our work should be Eqs. (4) and (5). We acknowledge that the exposition surrounding Eqs. (4) and (5) in the main text lacks clarity.

In essence, Eqs. (4) and (5) can be readily derived based on the action of fermionic partial transpose on Gaussian states, specifically the relation $(\rho_0[G])^{T_2^f} = \rho_0[G^{T_2^f}]$ following Eq. (3). After Trotter decomposition and Hubbard-Stratonovich transformation, the interacting density matrix is decomposed into a weighted sum of Gaussian operators, $\rho = \sum_{\mathbf{s}} P_{\mathbf{s}} \rho_{\mathbf{s}}$. It is crucial to note that the partial transpose operation is linear, allowing us to apply the partial transpose to each individual term in the sum: $\rho^{T_2^f} = \sum_{\mathbf{s}} P_{\mathbf{s}} (\rho_{\mathbf{s}})^{T_2^f} = \sum_{\mathbf{s}} P_{\mathbf{s}} \rho_0[G_{\mathbf{s}}^{T_2^f}]$. This directly leads to Eqs. (4) and (5). In short, we argue that the derivation of Eqs. (4) and (5) is straightforward, provided that one accepts the preliminary results concerning Gaussian states presented prior to these two equations.

In Section S3 of the supplementary materials, we proved Eqs. (4) and (5) in a slightly different manner. Unlike the main text, where we discussed Gaussian states and interacting fermionic states separately, we began with the decomposed form of ρ for interacting fermions, as outlined in Section S2, and presented a comprehensive and self-consistent derivation of Eqs. (4) and (5). This approach results in a seemingly lengthy derivation. However, most of the effort in Section S3 essentially mirrors what is necessary for Gaussian states. First, we established the relation $\Gamma = \tanh(-W/2)$ for arbitrary Gaussian states. Next, we proved the crucial relation $(\rho_0[\Gamma])^{T_2^f} = \rho_0[\Gamma^{T_2^f}]$ in the Majorana basis by utilizing Wick’s theorem.

To make the narrative more unified and coherent, we have thoroughly revised Section S3 in the supplementary materials. In the revised version, the proof of Eqs. (4) and (5) is modularized into two distinct parts: first addressing Gaussian states, and then extending to interacting fermionic states. We have also refined several sentences in the main text section titled “Partially Transposed Density Matrix Integrated into the DQMC Framework” when referencing the supplementary materials. With these revisions, readers should be clearly guided to the appropriate (sub)sections in the supplementary materials when consulting this part of the main text.

To enhance clarity around Eqs. (4) and (5) in the main text, we have add several sentences to emphasize the linearity of the partial transpose operation and the application of previous results concerning Gaussian states. We recognize that the specific form of the partition function is not essential for readers to grasp the main idea, so we have removed the sentence involving the notation $K_l[s]$. The revised sentences preceding Eq. (4) in the main text now read as follows: “Consequently, since the product of Gaussian states remains a Gaussian state up to a normalization factor, the interacting fermionic density matrix $\rho \sim e^{-\beta H}$ can be expressed as a weighted sum of Gaussian operators. Explicitly, we have $\rho = \sum_{\mathbf{s}} P_{\mathbf{s}} \rho_{\mathbf{s}}$, where $P_{\mathbf{s}}$ represents the weight of the configuration \mathbf{s} (see the supplementary materials for concrete expressions of specific models). By leveraging the linearity of the partial transpose, we can first individually compute the FPT for each Gaussian state $\rho_{\mathbf{s}}$. We then sum these results, weighted by their respective probabilities $P_{\mathbf{s}}$, to obtain the FPT of the entire density matrix ρ : ...”

Reviewer #2: 2. When pointing out the fact that the logarithmic correction to the area law appears at twice the transition temperature, authors write “The inclusion of the prefactor 2 aligns with the rank of the RNR under consideration, consistent with earlier discussion on the critical behaviour within replica approach.” and refer to the literature. Since this result is even highlighted in the abstract, it would strengthen the manuscript to include its explanation, rather than providing a brief statement and a reference to the literature.

Our reply and changes: We thank the reviewer for the suggestion. We agree that offering more detailed explanations is important for the readers to understand the results presented in Fig. 3. In general, the singularity arising from the finite-temperature transition is reflected in the temperature-dependent behavior of the

rank- n Rényi negativity ratio at temperature $T = nT_c$, rather than $T = T_c$, where T_c denotes the critical temperature. This fact can be attributed to the replica trick and can be elucidated by examining the definition of the Rényi negativity ratio in Eq. (9): $R_n \equiv \frac{1}{1-n} \ln \left\{ \frac{\text{Tr}[(\rho^{T_2^f})^n]}{\text{Tr}[\rho^n]} \right\}$. The denominator, $\text{Tr}[\rho^n] = \text{Tr}[e^{-n\beta H}] = \text{Tr}[e^{-\beta_{\text{eff}} H}]$, represents the partition function of a Gibbs state at an effective temperature $T_{\text{eff}} = T/n$. Consequently, the singular point at $T = T_c$ for the original partition function $\text{Tr}[e^{-\beta H}]$ is mapped to nT_c for the replicated partition function $\text{Tr}[e^{-n\beta H}]$.

To better explain the results, we have added two additional sentences following the one highlighted by the reviewer: “In general, the singularity of R_n is anticipated to occur at $T = nT_c$ with T_c being the physical transition temperature. This expectation arises because R_n , as indicated by the denominator in its definition in Eq. (9), i.e., $\text{Tr}[e^{-n\beta H}]$, effectively corresponds to a Gibbs state with an effective inverse temperature of $n\beta$.”

Reviewer #2: 3. In the main text, some quantities are either never introduced or are introduced too late. For example, the column vector “gamma” is never introduced, while the column vector “c” is introduced near the partition function, despite its earlier appearance in the density matrix of a Gaussian state. Additionally, there are some errors in the Supplemental Material. For example, H_U and not H_I is introduced in Eq. (S7), indices are mixed in Eq. (S9), et cetera.

Our reply and changes: We thank the reviewer for careful reading our manuscript and pointing out these issues. In the revised manuscript, we have made the following changes to address these concerns:

- The column vector \mathbf{c} is now introduced in the second paragraph of the section titled “Partially transposed density matrix integrated to DQMC framework”, where we first define Dirac operators. The new added sentence reads: “In the following discussion, we may use a column vector $\mathbf{c} = (c_{1,\uparrow}, \dots, c_{N,\uparrow}, c_{1,\downarrow}, \dots, c_{N,\downarrow})^T$ to compactly encapsulate all the fermionic operators.”
- The column vector $\boldsymbol{\gamma}$ is now introduced after Eq. (1) where we first use it to write a Gaussian state. Now the sentence reads: “Remarkably, under this definition, the partial transpose of a Gaussian state, denoted as $\rho_0 \sim e^{\frac{1}{4}\boldsymbol{\gamma}^T W \boldsymbol{\gamma}}$ with $\boldsymbol{\gamma} = (\gamma_{1,\uparrow}, \dots, \gamma_{2N,\uparrow}, \gamma_{1,\downarrow}, \dots, \gamma_{2N,\downarrow})^T$, retains its Gaussian nature. ”
- All instances of H_U in Eq. (S7) have been corrected to H_I .
- We have revised the Hubbard-Stratonovich transformation for spinless t - V model in Eq. (S9) to

$$e^{-\Delta\tau V \sum_{\langle jk \rangle} (n_j - \frac{1}{2})(n_k - \frac{1}{2})} = \sum_{\{s_{jk} = \pm 1\}} \left(\frac{1}{2} e^{-\frac{V\Delta\tau}{4}} \right) e^{\lambda \sum_{\langle jk \rangle} s_{jk} (c_j^\dagger c_k + c_k^\dagger c_j)}, \quad (\text{S9})$$

so as to ensure consistent indices.

Additionally, we have carefully reviewed the supplementary materials and corrected other typos and errors, which are shown in red font in the revised version.

Reviewer #2: *4. Authors may consider adding a short description of the Determinant Quantum Monte Carlo framework to the Supplemental Material. While the main expressions are provided in the main text, short explanation of the framework itself would be beneficial.*

Our reply and change: We thank the reviewer for the suggestion. The supplementary materials already contain a brief introduction to the basic formalism of the Determinant Quantum Monte Carlo (DQMC), covering both the finite-temperature version and the projective algorithm. As exhibited in Section S2, we provided the most relevant formulas to this work, namely the decomposed expressions for the partition functions (Eqs. (S10) and (S13)) and the expectation of generic operators (Eqs. (S11) and (S14)). These weighted sum can be sampled by utilizing the quantum Monte Carlo technique.

As suggested by the reviewer, we have further briefly described the practical framework of DQMC, including updates, propagation, measurements and numerical stabilization, in the revised supplementary materials. We believe that this inclusion will assist readers in better understanding the workflow of our simulations.

Response to Nature Communications Submission (Round 2)

Paper Title: Entanglement Rényi negativity of interacting fermions from quantum Monte Carlo simulations

Authors: Fo-Hong Wang and Xiao Yan Xu

Summary of changes made

To sum up, we have made the following changes:

1. As suggested by Reviewer 2, we have improved the main text by including brief descriptions of the Trotter decomposition and Hubbard-Stratonovich transformation procedures before Eqs. (5) and (6).
2. Tiny adjustments to the figure labels at the beginning of the Supplementary Material.

The following is a detailed response to the recommendations and criticisms from the reviewers. The reviewer comments are laid out in *italicized font* and our response is given in normal font.

Reply to Reviewer #1

Reviewer #1: *It is my belief that the papers published in Nature Communications are intended to represent important advances in specific fields together with significance for physics. Therefore, in my first review I emphasized this viewpoint rather than going through the formulae in the manuscript.*

Our reply: We appreciate the reviewer’s clarification regarding their perspective on publications in Nature Communications. However, we maintain that our work meets these criteria by introducing novel methodological approaches that facilitate the investigation of mixed-state quantum entanglement in strongly correlated fermionic systems through the lens of Rényi negativity. Furthermore, we encourage the reviewer to refer to the introduction section of our manuscript, which provides a comprehensive overview of the current state of pure- and mixed-state entanglement studies in quantum many-body systems and highlights the significant advancements our work contributes. Notably, this section does not include any formulae.

Reviewer #1: *Also, I don’t think it is a good idea to question the reviewer’s competence as “lack of logical sequence in his/her comments” or a “lack of familiarity with the realm...” as the reviewer doesn’t need to be exactly an expert in “quantum many-body computations” to be able to judge whether a paper represents an important advance or not.*

Our reply: We sincerely apologize if our previous response seemed to question the reviewer’s competence; that was not our intention. We aimed to convey our challenges in addressing the **reports**, as the comments were either overly general or inaccurate. We did not imply that the reviewer must be an expert in quantum many-body computations to evaluate the significance of our work. However, when evaluations lack reasonable arguments or factual basis, it becomes challenging for us to accept them and make improvements to our manuscript accordingly.

Reviewer #1: *Here I’ll leave aside the statement in author’s response that the entanglement entropy cannot characterize entanglement of mixed state, which is simply wrong. For example, a simple molecule is in a pure state at zero temperature but a single atom within it is in a mixed state and its entanglement can be quantified by entanglement entropy.*

Our reply: We thank the reviewer for raising this specific concern about the mixed-state entanglement. We would like to elucidate this fundamental point from two aspects:

(1) The claim that “the entanglement within mixed states can be quantified by entanglement entropy” is incorrect. This statement undermines the motivation and significance of not only our present work but also many prior studies on proposing, validating, and computing mixed-state entanglement measures (see references in the second and third paragraphs of our manuscript). The limitations of entanglement

entropy are well-documented and can be easily illustrated using basic quantum information theory. Pure states can contain only quantum correlation, while mixed states can exhibit both quantum and classical correlations. For a bipartition of system $A = A_1 \cup A_2$, a separable mixed state is defined as a convex combination of product states, i.e., $\rho = \sum_i p_i \rho_{A_1}^{(i)} \otimes \rho_{A_2}^{(i)}$, where $p_i \geq 0$, $\sum_i p_i = 1$, and $\rho_{A_1}^{(i)}$ and $\rho_{A_2}^{(i)}$ are the density matrices within subsystems A_1 and A_2 , respectively. The set of separable states is larger than that of product states. While separable states can contain classical correlations from mixing, they are not entangled. For instance, the 2-qubit mixed state $\rho = \frac{1}{2}(|00\rangle\langle 00| + |11\rangle\langle 11|)$ is separable but has non-zero entanglement entropy $S = -\text{Tr} \rho_A \ln \rho_A = \ln 2$, reflecting only classical correlations and not quantifying entanglement. Conversely, the entanglement negativity between the two qubits is zero (this holds for all separable states; see Ref. [13-18] regarding the positive partial transpose criteria and the proposal of entanglement negativity).

(2) Regarding the example provided by the reviewer, we find the phrase “its entanglement” somewhat ambiguous. If the reviewer referred to the entanglement between the single atom and the rest of the molecule, which can be quantified by their entanglement entropy, we fully concur. This represents a typical example of bipartite **pure-state** entanglement. However, if the reviewer addressed the entanglement within the internal degrees of freedom of the single atom, we contend that the entanglement entropy is not an appropriate measure, as discussed in (1).

Reviewer #1: *I'll not also comment on the author's statement that their methods are applicable to “realistic many-body systems” simply because the validity of Hubbard model they employ is by itself a rather crude approximation to the realistic systems as shown previously. Building a new methodology based on an oversimplified model which neglects the detailed electron-electron interactions including the long-range interactions, etc, cannot in my view be considered an important advance. Clearly, benchmarking against exact diagonalization does not help much in this sense.*

Our reply: We appreciate the criticism and acknowledge the reviewer's concerns regarding the applicability of our method to realistic many-body systems. We agree that both the Hubbard model and the t - V model are oversimplified toy models that fail to encapsulate the full complexity of real materials. Nonetheless, the novelty of our work should not be solely assessed from the aspect of model calculations. We would like to reiterate the important advance of our work in the following aspects:

(1) The primary contribution of our work lies in the integration of the fermionic partial transpose within the framework of DQMC, leveraging the notable theoretical insight that the fermionic partial transpose of Gaussian states remains Gaussian. Therefore, our method is quite general and is not “built on an oversimplified model,” as suggested by the reviewer.

(2) We used the Hubbard model and the t - V model as examples to demonstrate our method's effectiveness. Although these models are simplified, they capture essential physics of strongly correlated electron systems, are widely studied, and help explain phenomena like high- T_c superconductivity [R1]. Additionally, our method

can be applied to various sign-problem-free fermionic models using the same DQMC and incremental algorithms, including models with long-range interactions (e.g., see Ref. [R2] for the honeycomb lattice with Hubbard and long-range Coulomb interactions).

(3) Our work indeed only serves as a preliminary step towards the computation of mixed-state entanglement in realistic many-body systems. However, this initial endeavor holds significant importance, and its potential applicability to realistic systems is not merely a hollow assertion. As elaborated in the Outlook section of the manuscript, the subsequent technical approaches are diverse and practical.

- (i) The first approach involves extending our method to the continuous-time quantum Monte Carlo (CTQMC) algorithm, which can then be utilized as the impurity solver in dynamical mean-field theory (DMFT) calculations (see Ref. [69] for a review on the synergy between these two algorithms), thus facilitating the investigation of realistic systems. We can compute the Rényi negativity within the framework of two types of continuous-time algorithms: the interaction expansion algorithm (CT-INT) and the hybridization expansion solvers (CT-HYB).
 - (a) CT-INT: The rank-two Rényi negativity can be computed using the interaction expansion algorithm via the identity $\text{Tr}[(\rho^{T_2^f})^2] = \text{Tr}[(\rho X_2(\pi))^2]$. In this context, the disorder operator $X_2(\tau) \propto \prod_{j \in A_2, \sigma} (n_{i\sigma}(\tau) - \frac{1}{2})$ introduces additional interaction vertices specifically in the A_2 subregion. For the impurity solver, the subsystem A_2 may encompass the entire bath; however, it can be readily generalized to lattice models, such as the Hubbard model, and to general bipartition geometries.
 - (b) CT-HYB: The hybridization expansion separates the impurity's degrees of freedom from the bath, allowing us to trace out the bath to obtain the reduced density matrix $\rho_{\text{imp}} = \text{Tr}_{\text{bath}}(\rho)$. We first compute the reduced Green's function G_{impurity} from selected matrix elements of the total Green's function. The reduced density matrix, of dimension $2^{n_{\text{fr}}}$ (where n_{fr} is the number of spins and orbitals), is derived from Green's functions $\langle c_i c_j^\dagger \rangle$ and density-density correlation functions $\langle n_i n_j \rangle$ (see Ref. [58] for the case of two spinless fermions), which are easily measured in DQMC. The partial transpose can then be performed to obtain the logarithmic negativity or Rényi negativity.
- (ii) The second approach is to incorporate the Rényi negativity into the constrained-path auxiliary-field quantum Monte Carlo (CP-AFQMC) method (see Ref. [74-77]), which alleviates the sign problem to some extent. Its generalization, the phaseless AFQMC method, has been successfully applied to investigate realistic systems, including molecules and other materials [R3,R4].

[R1] B. Keimer, S. A. Kivelson, M. R. Norman, S. Uchida, and J. Zaanen, From quantum matter to high-temperature superconductivity in copper oxides, *Nature* 518, 7538 (2015).

- [R2] H.-K. Tang, I. Yudhistira, U. Chattopadhyay, M. Ulybyshev, P. Sengupta, F. F. Assaad, and S. Adam, Spectral functions of lattice fermions on the honeycomb lattice with Hubbard and long-range Coulomb interactions, *Phys. Rev. B* 110, 155120 (2024).
- [R3] M. Suewattana, W. Purwanto, S. Zhang, H. Krakauer, and E. J. Walter, Phaseless auxiliary-field quantum Monte Carlo calculations with plane waves and pseudopotentials: Applications to atoms and molecules, *Phys. Rev. B* 75, 245123 (2007).
- [R4] Simons Collaboration on the Many-Electron Problem et al., Towards the Solution of the Many-Electron Problem in Real Materials: Equation of State of the Hydrogen Chain with State-of-the-Art Many-Body Methods, *Phys. Rev. X* 7, 031059 (2017).

Reviewer #1: *I think that beyond the technicalities the authors really overestimate the significance of their work saying that it “holds interdisciplinary significance, as it bridges the gap between quantum information theory and condensed matter physics”. I don’t think so, for I did not see any “new insights into entanglement in strongly correlated systems” or “new avenues” throughout the paper. What I saw are just numerical advances in treating fermionic systems under finite temperature conditions but with no new consequences for physics, no new conclusions, etc. In both the Abstract and the Conclusions, the authors are just saying what they have done. I hope they will finally understand what my point is here... A clear presentation of the limitations of the model, along with clear claims of novelty with respect to new physics, is still missing. Having said all that, I still believe that simply presenting an advance in numerical techniques would not be sufficient to warrant publication in Nature Communications.*

Our reply: We appreciate the reviewer’s perspective and acknowledge their concerns regarding the physical implications of our work. However, we respectfully disagree with the assertion that we are “just saying what we have done in both the Abstract and the Conclusions” and that our work “has no new consequences for physics and no new conclusions”. We would like to clarify the two key points that the reviewer may have overlooked:

(1) “new avenues”: Our work presents a novel methodological framework for investigating mixed-state quantum entanglement in strongly correlated fermionic systems via Rényi negativity. We perform the inaugural large-scale numerical analysis of Rényi negativity in two interacting fermionic lattice models, revealing novel physical results that illustrate the quantum-classical crossover and finite-temperature phase transition through the lens of entanglement. This method is versatile and can be applied to various QMC algorithms, demonstrating significant potential for exploring realistic many-body systems.

(2) “new insights into entanglement in strongly correlated systems”: As stated in the abstract and conclusions, we demonstrate that the Rényi negativity ratio serves

as an effective indicator for detecting the finite-temperature phase transition in the t - V model. This highlights the **close relationship between entanglement properties and critical behaviors in strongly correlated systems**. Furthermore, the quantitative scaling behavior of the Rényi negativity ratio—specifically, the logarithmic divergence of the area law coefficient at the finite-temperature transition point—offers new insights into the **distinction between bosonic and fermionic entanglement**. Previous studies on the rank-3 Rényi negativity in a transverse-field Ising model, which undergoes a finite-temperature transition within the same universality class as the t - V model, exhibited a significantly weaker singularity (to be more detailed, it does not exceed area-law behavior, manifesting only as a singularity in the temperature derivative of the rank-3 Rényi negativity).

We also hope that the reviewer will finally understand that the claims of novelty in our work are substantiated by the results and discussions presented in the manuscript. Again, while we respect the reviewer's opinions, it is challenging for us to address feedback that lacks a foundation in reasonable arguments or factual evidence.

Reply to Reviewer #2

Reviewer #2: *I am satisfied with the modifications to the manuscript made by the authors. Moreover, I stand by my previous assessment that the manuscript is interesting and well-written and, as such, it can be published in Nature Communications. I have only one final suggestion.*

Our reply: We sincerely thank the reviewer for their positive assessment and continued support of our manuscript.

Reviewer #2: *I propose that, instead of writing “Nonetheless, within the framework of DQMC, after Trotter decomposition and Hubbard-Stratonovich (HS) decoupling (see Methods and the supplementary materials), the original two-particle interaction terms in the Hamiltonian H are replaced by fermion bilinears coupled with spacetime dependent auxiliary fields s .”, the authors could provide an example. This example could be drawn from the beginning of Supplemental Material S2A, starting with “At a finite temperature T , and assuming that the system of interest is in thermodynamic equilibrium (...)” and ending with “Here, $s(l)$ includes all the auxiliary fields at time slice l and N_c represents the number of coupling terms, which varies depending on the specific interactions and decoupled channels.”. Including this example would help visualize the appearance of fermion bilinears while simultaneously establishing a connection with the remainder of the paragraph “Consequently, since the product of Gaussian states remains a Gaussian state (...).”*

Our reply and changes: We thank the reviewer for this constructive suggestion. We agree that including a concrete example would make the presentation more accessible and clearer. As suggested, we have revised the manuscript to include detailed descriptions of the Trotter decomposition and Hubbard-Stratonovich transformation procedures in the main text, following the sentence “It is now pertinent to ... are not Gaussian states.” and preceding the sentence “By leveraging the linearity of the partial transpose ...”. The revised sentences are highlighted in red within the revised manuscript.